# Scaling the Codebook Size of VQGAN to 100,000 with a Utilization Rate of 99%

**Lei Zhu**[1]    **Fangyun Wei**[2*]    **Yanye Lu**[1]    **Dong Chen**[2]
[1]Peking University    [2]Microsoft Research Asia
zhulei@stu.pku.edu.cn    fawe@microsoft.com    yanye.lu@pku.edu.cn    doch@microsoft.com

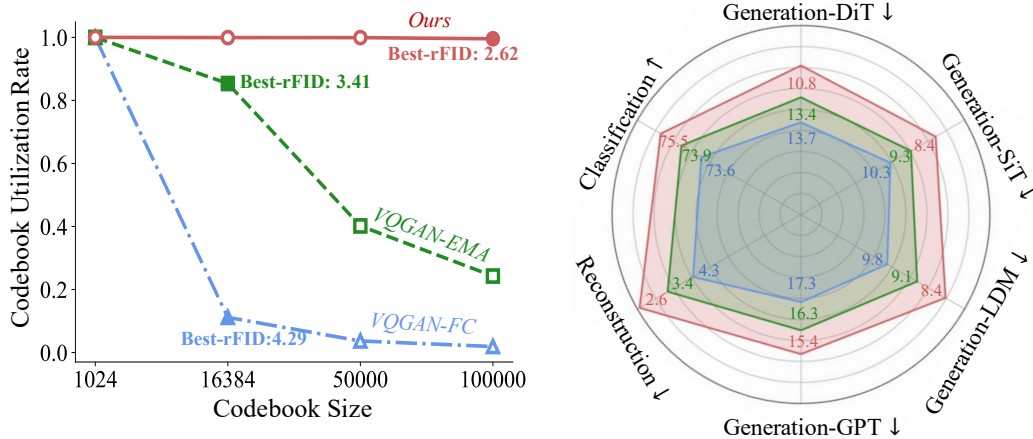

(a) Codebook size v.s. utilization rate.      (b) Evaluation on downstream tasks.

Figure 1: (a) Two enhanced versions of VQGAN [1], namely VQGAN-FC (Factorized Codes) and VQGAN-EMA (Exponential Moving Average), experience a decline in codebook utilization rate and performance as their codebook sizes expand. In contrast, our method, VQGAN-LC (Large Codebook), effectively leverages an extremely large codebook, persistently maintaining a utilization rate of up to 99% and achieving higher performance. We highlight the best reconstruction rFID for each model. (b) Comparison among three models across various tasks. For image generation, we evaluate the applications of these three VQGAN variants to GPT [2], LDM [3], DiT [4] and SiT [5].

## Abstract

In the realm of image quantization exemplified by VQGAN, the process encodes images into discrete tokens drawn from a codebook with a predefined size. Recent advancements, particularly with LLAMA 3, reveal that enlarging the codebook significantly enhances model performance. However, VQGAN and its derivatives, such as VQGAN-FC (Factorized Codes) and VQGAN-EMA, continue to grapple with challenges related to expanding the codebook size and enhancing codebook utilization. For instance, VQGAN-FC is restricted to learning a codebook with a maximum size of 16,384, maintaining a typically low utilization rate of less than 12% on ImageNet. In this work, we propose a novel image quantization model named VQGAN-LC (Large Codebook), which extends the codebook size to 100,000, achieving an utilization rate exceeding 99%. Unlike previous methods that optimize each codebook entry, our approach begins with a codebook initialized with 100,000 features extracted by a pre-trained vision encoder. Optimization then focuses on training a projector that aligns the entire codebook with the feature distributions of the encoder in VQGAN-LC. We demonstrate the superior performance of our model over its counterparts across a variety of tasks, including image reconstruction, image classification, auto-regressive image generation using GPT, and image creation with diffusion- and flow-based generative models.

---

*Corresponding author.

38th Conference on Neural Information Processing Systems (NeurIPS 2024).

Table 1: We conduct a comparative analysis of our VQGAN-LC against two advanced variants of VQGAN [1], namely VQGAN-FC and VQGAN-EMA, focusing on the effects of enlarging their codebook sizes from 1,024 to 100K. The only difference among the three models lies in the initialization and optimization of the codebook. The evaluation covers both reconstruction and generation using the latent diffusion model (LDM) [3] on the ImageNet dataset.

| Method | Reconstruction (rFID) | | | | Generation with LDM [3] (FID) | | | |
|---|---|---|---|---|---|---|---|---|
| | 1,024 | 16,384 | 50K | 100K | 1,024 | 16,384 | 50K | 100K |
| VQGAN-FC | 4.82 | **4.29** | 4.96 | 4.65 | 10.81 | **9.78** | 10.37 | 10.12 |
| VQGAN-EMA | 4.93 | **3.41** | 3.88 | 3.46 | 10.16 | **9.13** | 9.29 | 9.50 |
| VQGAN-LC (Ours) | 4.97 | 3.01 | 2.75 | **2.62** | 9.93 | 8.84 | 8.61 | **8.36** |

# 1 Introduction

Image quantization [1, 6, 7] refers to the process of encoding an image into a set of discrete representations, also known as image tokens, each derived from a codebook of a pre-defined size. VQGAN [1] stands out as a prominent architecture, with an encoder-quantizer-decoder structure, playing a pivotal role in various applications, including: (1) training a GPT [2, 8, 9, 10, 11] on image tokens to create images; (2) serving as an autoencoder in latent diffusion models (LDMs) [3, 4] and generative models [12, 13, 14], with flow matching [5, 15]; and (3) functioning within large multi-modality models [16, 17, 18, 19], where its encoder processes input images and its decoder assists in image generation.

In contrast to natural languages, which typically maintain a static vocabulary, image quantization models necessitate a codebook of a pre-defined size to convert images into discrete image tokens. The nature of image signals—complex and continuous—makes translating images into token maps a form of lossy compression that is generally more severe than converting them into continuous feature maps. The capability of these models to represent images largely depends on the codebook size. Previous studies, such as VQGAN [1], its improved versions, including VQGAN with exponential moving average (EMA) update (VQGAN-EMA) and VQGAN using factorized codes (VQGAN-FC), and its predecessors, like VQVAE [6] and VQVAE-2 [7], have demonstrated that they can only learn a codebook with a maximum size of 16,384. These models often face unstable training or performance saturation issues when the codebook size is further increased, as shown in Table 1. Additionally, they typically exhibit a low codebook utilization rate—for instance, under 12% in VQGAN-FC, as shown in Figure 1(a)—indicating that a significant portion of the codebook remains unused, thereby diminishing the model's representational capacity. Furthermore, studies on large language models suggest that employing a tokenizer with an expanded vocabulary significantly enhances model efficacy. For example, the technical report for LLAMA 3[2] shows, "LLAMA 3 uses a tokenizer with a vocabulary of 128K tokens that encodes language much more efficiently, which leads to substantially improved model performance."

In this study, we investigate the scalability of codebook size in VQGAN and the improvement of its codebook utilization rate, thereby substantially enhancing the representational capabilities of VQGAN. Typically, as shown in Figure 2(a), image quantization models like VQGAN are structured with an encoder-quantizer-decoder architecture, where the quantizer is connected to a codebook. For a given image, the encoder produces a feature map that the quantizer then converts into a token map. Each token in this map corresponds to an entry in the codebook, based on their cosine similarity. This token map is subsequently used by the decoder to reconstruct the original image.

Generally, the codebook in VQGAN begins with a *random* initialization. Each entry (a.k.a. a token embedding) in the codebook is designated as *trainable* and undergoes optimization through either gradient descent [6, 1, 20, 21] (Figure 2(b)) or an exponential moving average (EMA) update [7, 22] (Figure 2(c)) during the training phase. Nevertheless, in each iteration, only a small amount of token embeddings, corresponding to the token maps of the current training batch, are optimized. As training progresses, these frequently optimized token embeddings gradually align more closely with the distributions of the feature maps generated by the encoder, compared to those less frequently or never optimized (referred to as inactive token embeddings). Consequently, these inactive token embeddings

---

[2] https://ai.meta.com/blog/meta-llama-3/

are excluded from the training process and subsequently remain unused during the inference phase, resulting in poor codebook utilization.

Our approach deviates from conventional image quantization models by initiating with a codebook composed of $N$ *frozen* features, sourced from a pretrained image backbone like the CLIP-vision-encoder [23], and utilizing datasets like ImageNet [24]. A projector is then employed to transition the entire codebook into a latent space, producing token embeddings. During the training process, it is the projector that is optimized, not the codebook itself, which distinguishes our method from traditional models. By optimizing the projector, we adapt the aggregate distribution of the codebook entries to align with the feature maps generated by the encoder. This contrasts with methods like VQGAN [1], where adaptations are made to a limited number of codebook entries to match the feature map distributions during each iteration. Our simple quantization technique ensures that almost all token embeddings (over 99%) remain active throughout the training phase. The process is depicted in Figure 2(d).

Our newly developed quantizer can be integrated directly into the existing VQGAN architecture, replacing its standard quantizer without requiring any changes to the encoder and decoder. This innovative quantizer enables the expansion of the codebook to sizes up to 100,000, while maintaining an impressive utilization rate of 99%. By comparison, the conventional VQGAN is limited to a codebook size of 16,384 with a utilization rate of only 11.2% when applied to ImageNet. The advantages of a larger codebook with enhanced utilization are demonstrated across various applications, including image reconstruction, image classification, auto-regressive image generation using GPT, and image creation with diffusion models and flow matching. Figure 1 illustrates the performance of our improved VQGAN, termed VQGAN-LC (Large Codebook), compared to its counterparts.

## 2 Related Work

**Image Quantization.** Image quantization focuses on compressing an image into discrete tokens derived from a codebook [6, 1, 22, 20, 25, 26, 27]. VQVAE [6] introduces a method of quantizing patch-level features using the nearest codebook entry, with the codebook learned jointly with the encoder-decoder structure through reconstruction loss. VQVAE2 [7] enhances this by employing exponential moving average updates and a multi-scale hierarchical structure to improve quantization performance. VQGAN [1] further refines VQVAE by integrating adversarial and perceptual losses, enabling more accurate and detailed representations. ViT-VQGAN [21] replaces the CNN-based encoder-decoder [28] with a vision transformer and shows that using a factorized code mechanism with $l_2$ regularization can improve codebook utilization. Reg-VQ [29] introduces prior distribution regularization to prevent collapse and low codebook utilization. Additionally, some approaches use codebook reset strategies to reset unused codebook entries during training [26, 30, 31, 32] or utilizing stochastic quantization to enhance utilization rates [26, 29]. In contrast to these methods, the proposed VQGAN-LC initializes codebook entries using a pre-trained vision encoder on target datasets, ensuring nearly full codebook utilization throughout the training process and allowing for scaling up the codebook size to more than 100K.

**Tokenized Image Synthesis.** Substantial advancements have been achieved in the realm of image synthesis, particularly through image quantization techniques. Initial efforts such as PixelRNN [33] employs LSTM networks [34] to autoregressively model dependencies between quantized pixels. Building upon this, the groundbreaking VQVAE [6] introduces the quantization of image patches into discrete tokens, significantly enhancing generation capabilities when paired with PixelCNN [35]. The iGPT [36] further advances the field by leveraging the powerful Transformer [37] for sequence modeling of VQ-VAE tokens. Recently, there has been a shift towards using non-autoregressive Transformers for image synthesis [12, 13, 14, 11], which provide efficiency improvements over traditional raster-scan-based generation methods. Innovative approaches such as discrete diffusion models, including D3PMs [38] and VQ-Diffusion [39], utilize discrete diffusion processes to model the distribution of image tokens. Additional diffusion-based techniques [3, 15, 4, 5, 40, 41] compress images into latent representations using quantizers, thereby reducing both training and inference costs. Moreover, image quantizers can enhance large language models for both image synthesis and understanding [17, 16, 18, 42, 19]. Our work introduces a superior image quantizer, further refining the image synthesis process.

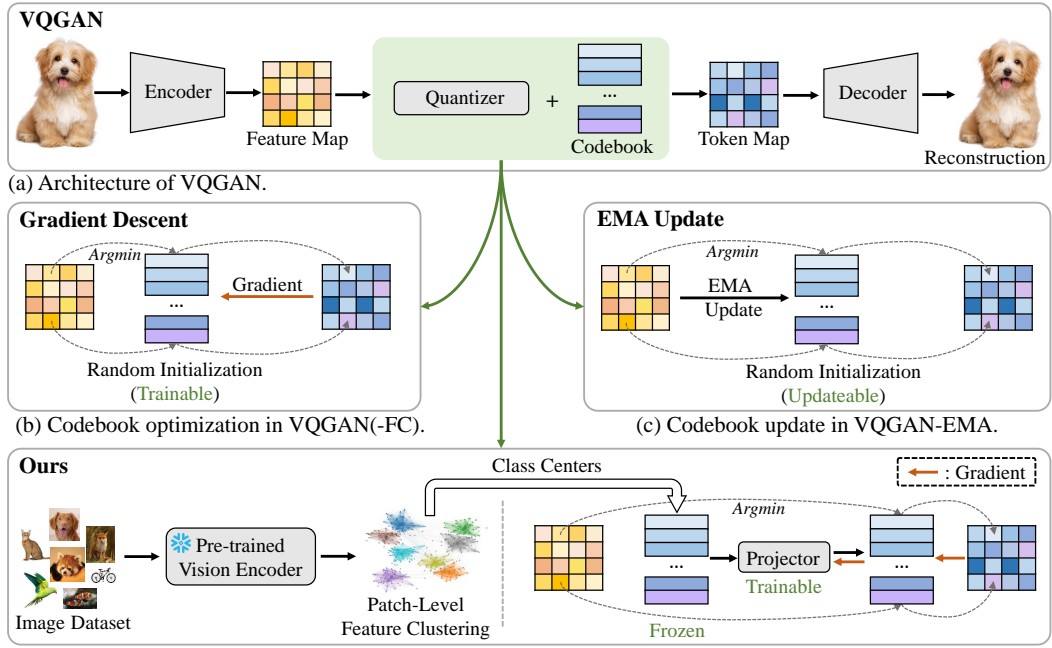

(a) Architecture of VQGAN.

(b) Codebook optimization in VQGAN(-FC).

(c) Codebook update in VQGAN-EMA.

(d) Codebook initialization and quantization process in our VQGAN-LC.

Figure 2: (a) The encoder-quantizer-decoder structure of VQGAN, with a codebook linked to the quantizer. (b) The codebook optimization strategy employed in VQGAN and VQGAN-FC. (c) The codebook update mechanism utilized in VQGAN-EMA. (d) The codebook initialization and quantization process implemented in our VQGAN-LC.

## 3 Method

### 3.1 Preliminary

**VQGAN.** Let $\mathcal{B} = \{\boldsymbol{b}_n \in \mathbb{R}^D\}_{n=1}^N$ denote a codebook containing $N$ entries, with each entry $\boldsymbol{b}_i$ being a $D$-dimensional *trainable* embedding with *random* initialization. As shown in Figure 2(a), VQ-GAN adopts an encoder-quantizer-decoder structure. In this setup, an encoder processes an image $\boldsymbol{X}$ of height $H$ and width $W$ to generate a feature map $\boldsymbol{Z} \in \mathbb{R}^{h \times w \times D}$, with $(h, w)$ representing the latent dimensions. Subsequently, the quantizer maps $\boldsymbol{Z}$ to a token map $\hat{\boldsymbol{Z}}$, where each token in $\hat{\boldsymbol{Z}}$ is an entry in $\mathcal{B}$ based on the cosine distance between $\boldsymbol{Z}$ and $\mathcal{B}$. Finally, the decoder reconstructs the original image from the token map $\hat{\boldsymbol{Z}}$. The entire network is optimized using a combination of losses, expressed as follows:

$$\mathcal{L} = \underbrace{||\hat{\boldsymbol{X}} - \boldsymbol{X}||^2}_{\mathcal{L}_R} + \underbrace{\alpha||sg(\hat{\boldsymbol{Z}}) - \boldsymbol{Z}|| + \beta||sg(\boldsymbol{Z}) - \hat{\boldsymbol{Z}}||}_{\mathcal{L}_Q} + \mathcal{L}_P + \mathcal{L}_{GAN}, \tag{1}$$

where $sg(\cdot)$ denotes the stop-gradient operation. The terms $\mathcal{L}_R$, $\mathcal{L}_Q$, $\mathcal{L}_P$ and $\mathcal{L}_{GAN}$ represent the reconstruction loss, quantization loss, VGG-based perceptual loss [1], and GAN loss [1], respectively. Hyper-parameters $\alpha$ and $\beta$ are set to 1.0 and 0.33 by default. As shown in Figure 2(b), we refer to the codebook optimization strategy used in the original VQGAN as "gradient descent".

**VQGAN-FC.** VQGAN faces significant challenges with inefficient codebook utilization. To address this issue, the factorized code (FC) mechanism, initially proposed by ViT-VQGAN [21], is employed. We refer to VQGAN integrated with the FC mechanism as VQGAN-FC. The key differences between VQGAN and VQGAN-FC are two-fold: 1) a linear layer is added to project the encoder feature $\boldsymbol{Z} \in \mathbb{R}^{h \times w \times D}$ into a low-dimensional feature $\boldsymbol{Z'} \in \mathbb{R}^{h \times w \times D'}$, where $D' \ll D$; 2) the codebook $\mathcal{B}$, consisting of $N$ $D'$-dimensional trainable embeddings, is randomly initialized. Consequently, the

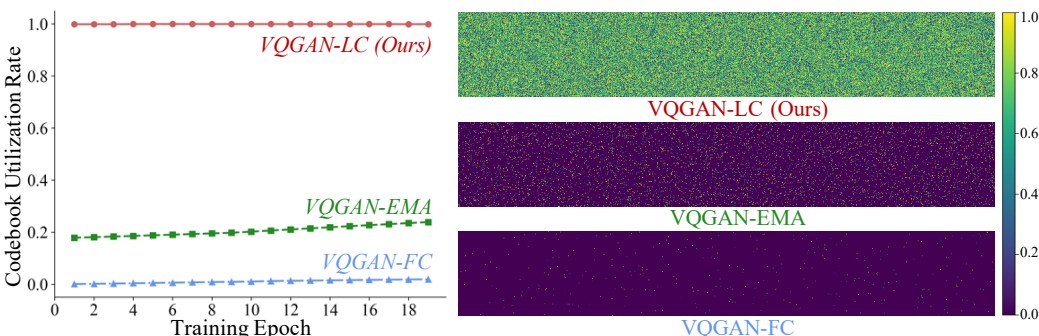

Figure 3: (Left) The codebook utilization rate over the training epoch. A codebook entry is considered utilized for the epoch if it is used at least once. (Right) The average utilization frequency of each codebook entry over all epochs, with each pixel representing a single entry. All models adopt a codebook with a size of 100K and use images with a resolution of $256 \times 256$ on ImageNet.

quantization loss in Eq. 1 is reformulated as:

$$\mathcal{L}_Q = \alpha||sg(\hat{\boldsymbol{Z}}) - \boldsymbol{Z}'|| + \beta||sg(\boldsymbol{Z}') - \hat{\boldsymbol{Z}}||. \tag{2}$$

However, as illustrated in Figure 1(a), the utilization rate of VQGAN-FC is only 11.2% on ImageNet when the codebook size is configured to 16,384, and increasing the size of the codebook fails to enhance performance as demonstrated in Table 1.

**VQGAN-EMA.** As depicted in Figure 2(c), this variant of VQGAN adopts an exponential moving average (EMA) strategy to optimize the codebook. Specifically, Let $\hat{\mathcal{B}} \subset \mathcal{B}$ denote the set of token embeddings used for all token maps in the current training batch. The set $\hat{\mathcal{B}}$ is updated through the EMA mechanism using the corresponding encoder features $\boldsymbol{Z}$ in each iteration. As a result, the codebook does not receive any gradients. Therefore, the quantization loss in Eq. 1 is defined as:

$$\mathcal{L}_Q = \alpha||sg(\hat{\boldsymbol{Z}}) - \boldsymbol{Z}||. \tag{3}$$

Our results, highlighted in Figure 1, indicate that VQGAN-EMA outperforms VQGAN-FC on various downstream tasks, leading to enhanced utilization of the codebook. However, expanding the codebook size continues to pose a significant challenge for VQGAN-EMA, as detailed in Table 1.

## 3.2 VQGAN-LC

**Analysis of VQGAN-FC and VQGAN-EMA**. In these enhanced versions of VQGAN, the codebook is initialized randomly. During each iteration, only a small subset of entries related to the current training batch are optimized. As a result, the frequently optimized entries become more aligned with the feature map distributions generated by the encoder, while the less frequently optimized entries remain underutilized. Consequently, a significant portion of the codebook remains unused during both the training and inference stages. Figure 3 shows the codebook utilization rate over the training epoch and visualizes the utilization frequency of each codebook entry once training is completed.

**Overview**. We present VQGAN-LC (Large Codebook), which allows for the expansion of the codebook to sizes of up to 100,000 while achieving a remarkable utilization rate of 99%. As illustrated in Figure 2(d), our method diverges from VQGAN-FC and VQGAN-EMA in its design of the quantizer. We maintain a static codebook and train a projector to map the entire codebook into a latent space, aligning the distributions of the feature maps generated by the encoder. This approach allows us to scale the codebook size effectively without modifying the encoder and decoder, achieving an extremely high utilization rate and resulting in superior performance across various tasks, as shown in Figure 1, Table 1 and Figure 3. It is important to note that *increasing the codebook size incurs almost no additional computational cost.*

**Codebook Initialization.** To initialize a static codebook, we first utilize a pre-trained vision encoder (e.g., CLIP with a ViT backbone) to extract patch-level features from the target dataset (e.g., ImageNet) containing $M$ images. This extraction results in a set of features denoted as $\mathcal{F} = \{\boldsymbol{F}_m^{(i,j)} \in$

$\mathbb{R}^D\}_{i=1,j=1,m=1}^{\bar{h},\bar{w},M}$, where $\boldsymbol{F}_m^{(i,j)}$ represents a $D$-dimensional patch-level feature at location $(i,j)$ in the $m$-th image, and $(\bar{h}, \bar{w})$ indicate the spatial dimensions of $\boldsymbol{F}$. Subsequently, we apply K-means clustering to $\mathcal{F}$, resulting in $N$ cluster centers (with a default value of $N = 100,000$). These cluster centers form the set $\mathcal{C} = \{\boldsymbol{c}_n \in \mathbb{R}^D\}_{n=1}^N$, where $\boldsymbol{c}_n$ is the $n$-th center. Our codebook $\mathcal{B}$ is then initialized using $\mathcal{C}$.

**Quantization.** Unlike VQGAN, VQGAN-FC and VQGAN-EMA, which optimize the codebook directly, our approach involves training a projector $P(\cdot)$, implemented as a simple linear layer, to align the static codebook $\mathcal{B}$ with the feature distributions generated by the encoder $E(\cdot)$ of our VQGAN-LC. Let $\mathcal{B}' = P(\mathcal{B}) = \{\boldsymbol{b}_n' \in \mathbb{R}^{D'}\}_{n=1}^N$ denote the projected codebook. For a given input image $\boldsymbol{X}$, the quantizer transforms the feature map $\boldsymbol{Z} = E(\boldsymbol{X}) \in \mathbb{R}^{h \times w \times D'}$ into a token map $\hat{\boldsymbol{Z}}$. This quantization process can be expressed as $\hat{\boldsymbol{Z}} := \underset{\boldsymbol{b}_n' \in \mathcal{B}'}{\mathrm{argmin}} ||\boldsymbol{Z}^{(i,j)} - \boldsymbol{b}_n'||$.

**Loss Function.** We employ the same loss function as specified in Eq.1. However, the key distinction is that our codebook $\mathcal{B}$ remains frozen, while the newly introduced projector $P(\cdot)$ undergoes optimization.

### 3.3 Evaluation of Image Quantization Models

We evaluate the performance of VQGAN-FC, VQGAN-EMA, and our proposed VQGAN-LC across image reconstruction, image classification and image generation tasks.

**Image Reconstruction.** Images are processed through the encoder, quantizer, and decoder to produce reconstructed images. These reconstructed images are then compared to their original images using the rFID metric as the evaluation criterion.

**Image Classification.** Initially, the encoder and quantizer convert each image into a token map. Subsequently, we utilize a ViT-B model [37], pre-trained with MAE [43], to train on all token maps for the purpose of image classification. Top-1 accuracy is used as the evaluation metric.

**Image Generation.** Image quantization models can be integrated with different image generation frameworks, such as auto-regressive causal Transformers (GPT [2]), latent diffusion models (LDM [3]), diffusion Transformers (DiT [4]), and flow-based generative models (SiT [5]), to facilitate image creation.

*GPT.* The encoder and quantizer transform each image into a token map $\hat{\boldsymbol{Z}}$, which is then flattened into a token sequence. Ultimately, GPT is trained on the collection of these token sequences.

*LDM.* It progressively adds noise onto the encoder feature $\boldsymbol{Z}$. The training objective is to denoise and reconstruct $\boldsymbol{Z}$. During the inference phase, the output from LDM is inputted into the quantizer and decoder of image quantization models to generate images.

*DiT.* This model is a variant of LDM, distinguished by its use of a Transformer architecture as the backbone. The incorporation of image quantization models into DiT follows the same approach as their integration into LDM.

*SiT.* This method presents a flow-based generative framework utilizing the DiT backbone. The integration of image quantization models in SiT follows the same methodology as in LDM and DiT.

## 4 Experiments

### 4.1 Setup

**Implementation Details of Image Quantization.** All image quantization models, including VQGAN, VQGAN-FC, VQGAN-EMA, and our proposed VQGAN-LC, utilize the same encoder and decoder of the original VQGAN. The input images are processed at a resolution of $256 \times 256$ pixels. The encoder (U-Net [28]) downsamples the input image by a factor of 16, yielding a feature map $\boldsymbol{Z}$ with dimensions of $16 \times 16$. The quantizer then converts this feature map into a token map $\hat{\boldsymbol{Z}}$ of the same size, which is subsequently fed into the decoder (U-Net) for image reconstruction. In our observations, the optimal codebook size for VQGAN, VQGAN-FC, and VQGAN-EMA is 16,384, whereas for our VQGAN-LC, the optimal codebook size is 100,000. Training is conducted on the

Table 2: Reconstruction performance on ImageNet-1K. The term "# Tokens" refers to the number of tokens used to represent an image. The codebook utilization rate is computed across all training images. The FC and EMA mechanisms are originally introduced by ViT-VQGAN [21] and VQVAE [6, 7], respectively. It is important to note that *increasing the codebook size incurs almost no additional computational cost.*

| Method | # Tokens | Codebook Size | Utilization (%) | rFID | LPIPS | PSNR | SSIM |
|---|---|---|---|---|---|---|---|
| DQVAE [20] | 256 | 1,024 | - | 4.08 | - | - | - |
| DF-VQGAN [46] | 256 | 12,288 | - | 5.16 | - | - | - |
| DiVAE [47] | 256 | 16,384 | - | 4.07 | - | - | - |
| RQVAE [22] | 256 | 16,384 | - | 3.20 | - | - | - |
| RQVAE [22] | 512 | 16,384 | - | 2.69 | - | - | - |
| RQVAE [22] | 1,024 | 16,384 | - | 1.83 | - | - | - |
| DF-VQGAN [46] | 1,024 | 8,192 | - | 1.38 | - | - | - |
| | 256 | 16,384 | 3.4 | 5.96 | 0.17 | 23.3 | 52.4 |
| VQGAN [1] | 256 | 50,000 | 1.1 | 5.44 | 0.17 | 22.5 | 52.5 |
| | 256 | 100,000 | 0.5 | 5.44 | 0.17 | 22.3 | 52.5 |
| | 256 | 16,384 | 11.2 | 4.29 | 0.17 | 22.8 | 54.5 |
| VQGAN-FC [21] | 256 | 50,000 | 3.6 | 4.96 | 0.15 | 23.1 | 54.7 |
| | 256 | 100,000 | 1.9 | 4.65 | 0.15 | 22.9 | 55.1 |
| | 256 | 16,384 | 83.2 | 3.41 | 0.14 | 23.5 | 56.6 |
| VQGAN-EMA [7] | 256 | 50,000 | 40.2 | 3.88 | 0.14 | 23.2 | 55.9 |
| | 256 | 100,000 | 24.2 | 3.46 | 0.13 | 23.4 | 56.2 |
| | 256 | 16,384 | **99.9** | 3.01 | 0.13 | 23.2 | 56.4 |
| VQGAN-LC (Ours) | 256 | 50,000 | **99.9** | 2.75 | 0.13 | 23.8 | 58.4 |
| | 256 | 100,000 | **99.9** | 2.62 | 0.12 | 23.8 | 58.9 |
| | 1,024 | 100,000 | 99.5 | **1.29** | **0.07** | **27.0** | **71.6** |

ImageNet-1K [24] and FFHQ [44] datasets, utilizing 32 Nvidia V100 GPUs. For ImageNet-1K, we train for 20 epochs, whereas for FFHQ, we train for 800 epochs. The Adam optimizer [45] is used, starting with an initial learning rate of $5e^{-4}$. This learning rate follows a half-cycle cosine decay schedule after a linear warm-up phase of 5 epochs.

**Codebook Initialization of Our VQGAN-LC.** Unless otherwise specified, we use the CLIP image encoder [23] with a ViT-L/14 backbone, adding an additional $4 \times 4$ average pooling layer, to extract patch-level features from images in the training split of the target dataset (either ImageNet or FFHQ). These features are then clustered into $N$ groups using the K-Means algorithm with CUDA acceleration. The cluster centers constitute the codebook. By default, $N$ is configured to 100,000. We specify the codebook entries to have a dimension of 8.

**Image Generation Models.** For LDM [3], DiT [4] and SiT [5], we adopt their original architectures. For generation using GPT [2], we follow VQGAN [1], using a causal Transformer decoder with 24 layers, 16 heads per attention layer, a latent dimension of 1,024 and a total of 404M parameters. For ImageNet, we employ class-conditional generation, whereas for FFHQ, we use unconditional generation. In LDM, DiT, and SiT, classifier-free guidance [3] is implemented for class-conditional generation. More implementation details can be found in Section A.

**Evaluation.** In the image reconstruction task, we evaluate performance using rFID, LPIPS, PSNR, and SSIM metrics on the validation sets of ImageNet and FFHQ. For image classification, we measure the top-1 accuracy on ImageNet. For image generation, we calculate the FID score on ImageNet using 50K generated images compared against the ImageNet training set. For FFHQ, the FID score is determined using 50K generated images in comparison with the combined training and validation sets of FFHQ. The codebook utilization rate is also reported for comparison, which is calculated as the ratio of active entries (tokens/codes) to the total size of the codebook.

## 4.2 Main Results

**Image Reconstruction.** Tables 2 and 3 present the reconstruction performance for ImageNet and FFHQ, respectively. We make three key observations: 1) Our method consistently achieves a

Table 3: Reconstruction performance on FFHQ.

| Method | # Tokens | Codebook Size | Utilization (%) | rFID | LPIPS | PSNR | SSIM |
|---|---|---|---|---|---|---|---|
| RQVAE [22] | 256 | 2,048 | - | 7.04 | 0.13 | 22.9 | 67.0 |
| VQWAE [48] | 256 | 1,024 | - | 4.20 | 0.12 | 22.5 | 66.5 |
| MQVAE [49] | 256 | 1,024 | 78.2 | 4.55 | - | - | - |
| VQGAN [1] | 256 | 16,384 | 2.3 | 5.25 | 0.12 | 24.4 | 63.3 |
| VQGAN-FC [21] | 256 | 16,384 | 10.9 | 4.86 | 0.11 | 24.8 | 64.6 |
| VQGAN-EMA [7] | 256 | 16,384 | 68.2 | 4.79 | 0.10 | 25.4 | 66.1 |
| VQGAN-LC (Ours) | 256 | 100,000 | **99.5** | **3.81** | **0.08** | **26.1** | **69.4** |

Table 4: Image generation on ImageNet-1K.

| Method | # Tokens | Codebook Size | Utilization (%) | FID |
|---|---|---|---|---|
| RQTransformer *(GPT-480M)* [22] | 256 | 16,384 | - | 15.7 |
| ViT-VQGAN *(GPT-650M)* [21] | 256 | 8,192 | - | 11.2 |
| DQTransformer *(GPT-355M)* [20] | 640 | 1,024 | - | 7.34 |
| DQTransformer *(GPT-655M)* [20] | 640 | 1,024 | - | 5.11 |
| ViT-VQGAN *(GPT-650M)* [21] | 1,024 | 8,192 | - | 8.81 |
| Stackformer *(GPT-651M)* [49] | 1,024 | 1,024 | - | 6.04 |
| LDM [3] | 1,024 | 16,384 | - | 8.11 |
| *with GPT-404M [2]* | | | | |
| VQGAN-FC [21] | 256 | 16,384 | 11.2 | 17.3 |
| VQGAN-EMA [7] | 256 | 16,384 | 83.1 | 16.3 |
| VQGAN-LC (Ours) | 256 | 16,384 | 99.9 | 16.1 |
| VQGAN-LC (Ours) | 256 | 100,000 | 97.0 | 15.4 |
| *with SiT-XL [5]* | | | | |
| VQGAN-FC [21] | 256 | 16,384 | 11.2 | 10.3 |
| VQGAN-EMA [7] | 256 | 16,384 | 83.1 | 9.31 |
| VQGAN-LC (Ours) | 256 | 16,384 | 99.9 | 9.06 |
| VQGAN-LC (Ours) | 256 | 100,000 | 99.6 | 8.40 |
| *with DiT-XL [4]* | | | | |
| VQGAN-FC [21] | 256 | 16,384 | 11.2 | 13.7 |
| VQGAN-EMA [7] | 256 | 16,384 | 85.3 | 13.4 |
| VQGAN-LC (Ours) | 256 | 16,384 | 99.9 | 11.2 |
| VQGAN-LC (Ours) | 256 | 100,000 | 99.4 | 10.8 |
| *with LDM [3]* | | | | |
| VQGAN-FC [21] | 256 | 16,384 | 11.2 | 9.78 |
| VQGAN-EMA [7] | 256 | 16,384 | 83.1 | 9.13 |
| VQGAN-LC (Ours) | 256 | 16,384 | 99.9 | 8.36 |
| VQGAN-LC (Ours) | 256 | 100,000 | 99.4 | 8.36 |
| VQGAN-LC (Ours) | 1,024 | 100,000 | 99.4 | **4.81** |

codebook utilization rate of over 99% across all codebook sizes on both datasets. 2) The reconstruction performance improves consistently with the scaling of codebook size using our method. 3) Increasing the codebook size (e.g., VQGAN-LC with a codebook size of 100,000 and 256 tokens), and the number of tokens to represent an image (e.g., RQVAE with a codebook size of 16,384 and 512 tokens) both enhance performance, with the former introducing almost no additional computational cost compared to the latter.

**Image Generation.** Table 4 shows the results of class-conditional image generation on ImageNet. All models (GPT, LDM, DiT, and SiT) demonstrate improved performance with the integration of our VQGAN-LC, regardless of their underlying architectures, which include auto-regressive causal Transformers, diffusion models, diffusion models with Transformer backbones, and flow-based generative models. The diversity of the generated images increases due to the utilization of a large codebook, which has a size of up to 100,000 and a utilization rate exceeding 99%. Table 5 displays the unconditional generation results on the FFHQ dataset. Notably, DiT and SiT, which use the Transformer architecture, require more extensive training data for optimizing diffusion- and flow-

Table 5: Image generation on FFHQ.

| Method | # Tokens | Codebook Size | Utilization (%) | FID |
|---|---|---|---|---|
| Stackformer *(GPT-307M)* [49] | 256 | 1,024 | - | 7.67 |
| DQTransformer *(GPT-308M)* [20] | 640 | 1,024 | - | 4.91 |
| Stackformer *(GPT-307M)*[49] | 1,024 | 1,024 | - | 6.84 |
| Stackformer *(GPT-651M)* [49] | 1,024 | 1,024 | - | 5.67 |
| ViT-VQGAN *(GPT-650M)* [21] | 1,024 | 8,192 | - | 3.13 |
| LDM [3] | 4,096 | 8,192 | - | 4.98 |
| *with LDM [3]* | | | | |
| VQGAN-FC | 256 | 16,384 | 11.2 | 13.2 |
| VQGAN-EMA | 256 | 16,384 | 68.2 | 12.5 |
| VQGAN-LC (Ours) | 256 | 100,000 | 99.7 | 12.3 |
| *with GPT (404M) [2]* | | | | |
| VQGAN-FC | 256 | 16,384 | 10.9 | 3.23 |
| VQGAN-EMA | 256 | 16,384 | 68.2 | 4.87 |
| VQGAN-LC (Ours) | 256 | 100,000 | 99.1 | **2.61** |

Table 6: Ablation study of using various codebook initialization strategies on ImageNet.

| Strategy | Dataset | Model | Utilization (%) | rFID | LPIPS | PSNR | SSIM |
|---|---|---|---|---|---|---|---|
| Random Initialization | - | - | 5.4 | 108.7 | 0.46 | 18.2 | 36.4 |
| Random Selection | ImageNet | ViT-L | 99.8 | 2.95 | **0.12** | **23.8** | **58.9** |
| K-Means Clutering | ImageNet | ResNet-50 | 99.8 | 2.71 | **0.12** | 23.7 | 58.3 |
| K-Means Clutering | ImageNet | ViT-B | **99.9** | 2.70 | **0.12** | **23.8** | 58.7 |
| K-Means Clutering | ImageNet | ViT-L | **99.9** | **2.62** | **0.12** | **23.8** | **58.9** |

based generative models. Given that FFHQ is significantly smaller than ImageNet, we limit our training on FFHQ to GPT and LDM.

**Image Classification.** In Section 3.3, we discuss the training of an image classifier on a dataset containing tokenized images. We fine-tune three ViT-B [37] models, pre-trained by MAE [43], using the tokenized images produced by the top-performing VQGAN-FC, VQGAN-EMA, and our proposed VQGAN-LC, on ImageNet. Both VQGAN-FC and VQGAN-EMA demonstrate optimal reconstruction performance when utilizing a codebook with 16,384 entries. As illustrated in Figure 1(b), our method achieves a top-1 accuracy of 75.7 on ImageNet, surpassing VQGAN-FC and VQGAN-EMA by margins of 1.6 and 1.9, respectively.

**Visualizations.** Section C presents images generated by GPT [2], LDM [3], DiT [4], and SiT [5], incorporating our VQGAN-LC.

## 4.3 Ablation Studies

Unless otherwise specified, we evaluate reconstruction performance on ImageNet across all studies.

**Codebook Initialization.** In Section 3.2, we describe the default codebook initialization approach. This involves using a CLIP vision encoder with a ViT-L backbone to extract patch-level features from ImageNet, followed by a K-means clustering algorithm to generate $N$ cluster centers, resulting in the static codebook $\mathcal{B}$. In Table 6, we evaluate two factors: 1) employing a CLIP vision encoder with different backbones (e.g., ViT-B [37] and ResNet-50 [50]) to extract patch-level features; and 2) utilizing non-clustering strategies to initialize the static codebook, including random initialization and random selection, where $N$ features are randomly chosen from all patch-level features of ImageNet. Our findings are threefold: 1) random initialization leads to extremely poor performance since the codebook remains frozen in our VQGAN-LC; 2) using a CLIP vision encoder with a ViT-L backbone outperforms those using ViT-B and ResNet-50 backbones; and 3) K-means clustering produces a more robust codebook.

**Codebook Size.** In Table 7, we incrementally increase the codebook size in our VQGAN-LC from 1,000 to 200,000. The performance shows minimal improvements beyond a codebook size of

Table 7: Ablation study of using different codebook sizes on ImageNet.

| Codebook Size | Utilization (%) | rFID | LPIPS | PSNR | SSIM |
|---|---|---|---|---|---|
| 1,000 | **100.0** | 4.98 | 0.17 | 22.9 | 55.3 |
| 10,000 | 99.8 | 3.80 | 0.14 | 23.3 | 57.2 |
| 50,000 | 99.9 | 2.75 | 0.13 | 23.8 | 58.4 |
| 100,000 | 99.9 | **2.62** | **0.12** | 23.8 | 58.9 |
| 200,000 | 99.8 | 2.66 | **0.12** | **23.9** | **59.2** |

Table 8: The computational cost of VQGAN-LC with different codebook sizes.

| Codebook Size | MACs | Model Size |
|---|---|---|
| 16,384 | 195.08G | 71.71M |
| 100,000 | 195.70G | 71.72M |

Table 9: Ablation study on codebook transferability. The term "source dataset→target dataset" indicates that the codebook is initialized using the source dataset, while our VQGAN-LC is trained on the target dataset.

| Setting | Utilization (%) | rFID | LPIPS | PSNR | SSIM |
|---|---|---|---|---|---|
| FFHQ→FFHQ | **99.5** | **3.81** | **0.08** | **26.1** | **69.4** |
| ImageNet→FFHQ | 99.4 | 4.08 | **0.08** | **26.1** | **69.4** |
| ImageNet→ImageNet | **99.9** | **2.62** | **0.12** | **23.8** | **58.9** |
| FFHQ→ImageNet | **99.9** | 2.91 | **0.12** | **23.8** | **58.9** |

100,000—specifically, only 0.01 PSNR and 0.03 SSIM gains are observed, when the codebook size reaches 200,000. Therefore, we consistently use a codebook size of 100,000 in all experiments. The codebook utilization rate consistently exceeds 99% across all configurations.

**Computational Cost.** The primary inference cost of VQ-GAN is attributed to the encoder and decoder. Increasing the codebook size $N$ incurs minimal additional cost since the matrix multiplication between $F$ and $B$ is negligible compared to the encoder and decoder processing time. Table 8, we present the multiply-accumulates (MACs) and model sizes for our VQGAN-LC models with codebook sizes of 16,384 and 100,000, respectively, when inferring an image of size $256 \times 256$.

**Codebook Transferability.** In our standard setup, both the codebook initialization and the VQGAN-LC training are conducted using the same dataset, either ImageNet or FFHQ. In Table 9, we examine the transferability of the codebook by initializing it with one dataset and training our VQGAN-LC on a different dataset. Our findings indicate that our method exhibits significant codebook transferability, highlighting the robustness of our codebook initialization process.

## 5 Conclusion

In this work, we introduce a novel image quantization model, VQGAN-LC, which extends the codebook size to 100,000, achieving a utilization rate exceeding 99%. Our approach significantly outperforms prior models like VQGAN, VQGAN-FC and VQGAN-EMA, across various tasks including image reconstruction, image classification and image synthesis using numerous generation models such as auto-regressive causal Transformers (GPT), latent diffusion models (LDM), diffusion Transformers (DiT), and flow-based generative models (SiT), while incurring almost no additional costs. Extensive experiments on ImageNet and FFHQ verify the effectiveness of our approach.

## 6 Acknowledgements

This work was supported in part by the Natural Science Foundation of China under Grant 623B2001, Grant 62394311, and Grant 82371112; in part by Beijing Natural Science Foundation under Grant Z210008; and in part by the High-Grade, Precision and Advanced University Discipline Construction Project of Beijing under Grant BMU2024GJJXK004.

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

Table 10: The impact of maintaining a static codebook and incorporating a projector on ImageNet.

| Static | Projector | Utilization (%) | rFID | LPIPS | PSNR | SSIM |
|---|---|---|---|---|---|---|
| ✓ | | 3.9 | 18.6 | 0.36 | 19.2 | 40.4 |
| | ✓ | 99.1 | 2.65 | 0.12 | 23.7 | 58.0 |
| ✓ | ✓ | **99.9** | **2.62** | **0.12** | **23.8** | **58.1** |

Table 11: Ablation study on the dimension of the projected codebook on ImageNet.

| Dimension | Utilization (%) | rFID | LPIPS | PSNR | SSIM |
|---|---|---|---|---|---|
| 8 | 99.8 | 2.66 | 0.12 | 23.9 | 59.2 |
| 16 | 99.8 | 2.36 | 0.12 | 23.8 | 58.8 |
| 32 | 99.8 | 2.75 | 0.13 | 23.6 | 58.7 |
| 128 | 99.8 | 2.37 | 0.12 | 23.6 | 58.3 |
| 256 | 99.8 | 2.49 | 0.12 | 23.9 | 59.4 |
| 512 | 99.9 | 2.76 | 0.12 | 23.5 | 58.1 |

## A    More Implementation Details

**GPT.** We train GPT with a batch size of 1024 across 32 Nvidia V100 GPUs. The Adam optimizer is employed with an initial learning rate of $4.5e^{-4}$, and the model is trained for 100 epochs. Moreover, a linear decay schedule is used for adjusting the learning rate. A 5-epoch linear warm-up phase is adopted. Top-k sampling is adopted for auto-regressive generation, where $k$ is set as 10% of the vocabulary size.

**LDM.** The implementation utilizes four UNet layers with channel dimensions of $\{256, 1024, 1024, 256\}$. Conditions are integrated through a cross-attention mechanism at each UNet layer. The model is trained using the Adam optimizer with an initial learning rate of $4.5e^{-4}$ and a batch size of 448, distributed across 8 Nvidia V100 GPUs. The training process is conducted over 100 epochs. The classifier-free guidance scale is set as 1.4 for class-conditional generation.

**DiT.** We employ the 28-layer DiT-XL model with a patch size of 2, consisting of 675 million parameters. The model features 16 attention heads and an embedding dimension of 1152. To handle class conditions, we utilize the AdaLN-Zero block. For optimization, the Adam optimizer is used with an initial learning rate of $4.5e^{-4}$, and the model is trained for 400,000 iterations on the ImageNet dataset. The training is performed with a batch size of 256 across 8 Nvidia V100 GPUs. The classifier-free guidance scale is set as 8 for class-conditional generation.

**SiT.** We utilize SiT-XL as our flow-based generative model, mirroring the architecture of DiT-XL. For optimization, the Adam optimizer is employed with an initial learning rate of $4.5e^{-4}$. The model undergoes training for 400,000 iterations on the ImageNet dataset. The training process is conducted with a batch size of 256, distributed across 8 Nvidia V100 GPUs. The classifier-free guidance scale is set to 8 for class-conditional generation.

## B    More Experiments

**Projector and Static Codebook.** As described in Section 3.2, the codebook is initialized using a CLIP vision encoder to extract patch-level features on ImageNet. During the training of our VQGAN-LC, we optimize a projector to map the entire codebook to a latent space. Table 10 illustrates the significance of tuning the projector on the static codebook by comparing our default strategy with two alternatives: 1) omitting the projector; and 2) making each entry in the initialized codebook trainable. Our results show that incorporating the projector markedly improves performance, whereas making the codebook entries trainable has minimal impact.

**Dimension of the Projected Codebook.** In our VQGAN-LC, the projector transforms the codebook $\mathcal{B}$ into a latent codebook $\mathcal{B}'$, with each entry in $\mathcal{B}'$ having a dimension denoted as $D'$. Table 11 shows the results of varying $D'$ from 8 to 512. We find that our model's performance remains stable regardless of the value of $D'$, consistently achieving a codebook utilization rate of over 99% in all configurations.

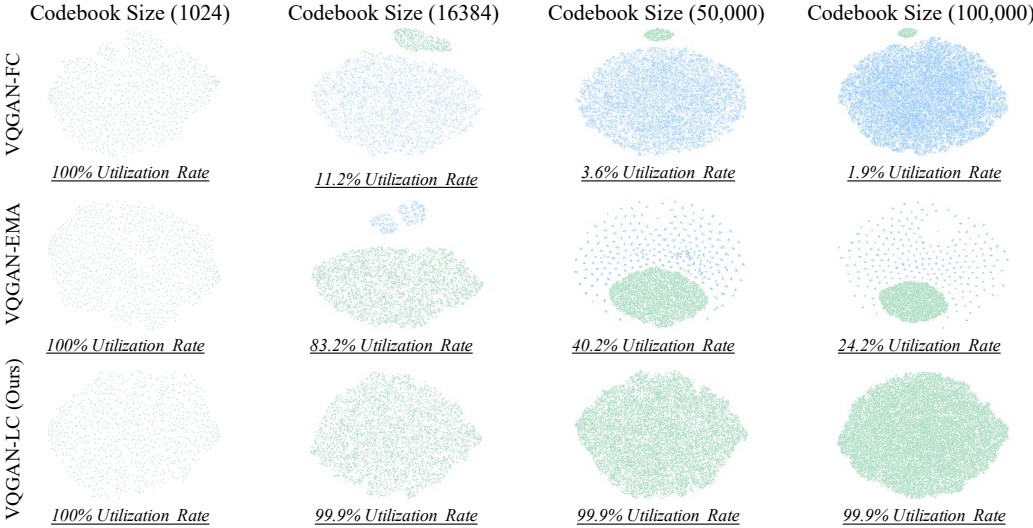

Figure 4: Visualization of the active and inactive codes for three models (VQGAN-FC, VQGAN-EMA, and our VQGAN-LC) using t-SNE.

**Visualization of Active and Inactive Codes.** Figure 4 shows the distribution of the active and inactive codes for three models: VQGAN-FC, VQGAN-EMA, and our VQGAN-LC. The visualization is created using t-SNE. Active codebook entries are highlighted in green, while inactive ones are shown in blue. As the codebook size increases, more codes tend to be inactive in both VQGAN-FC and VQGAN-EMA models.

**Token Replacement.** VQGAN, VQGAN-FC, and VQGAN-EMA all utilize a codebook of 16,384 entries to achieve optimal performance. In contrast, our VQGAN-LC can scale the codebook size up to 100,000 while maintaining an exceptionally high utilization rate of over 99%, allowing each token to represent more detailed visual elements. This is verified through an ablation study: for each input image, we use VQGAN-FC (-EMA/-LC)'s encoder to convert the image into a token map. We then replace each token in the map with the $M^{th}$ nearest entry from its codebook. The modified token map is fed into the decoder for reconstruction. Figure 5 shows the PSNR results on a subset of ImageNet, using images from 100 randomly selected categories, and visualizes the results of our VQGAN-LC, VQGAN-FC, and VQGAN-EMA when $M$ is set to 1, 50, 100, and 1000.

## C  Visualizations

In Figures 6-9, we present the class-conditional generation results at a resolution of $256 \times 256$ for our VQGAN-LC with GPT [2], LDM [3], DiT [4], and SiT [5], respectively, using 256 ($16 \times 16$) tokens on ImageNet. Additionally, Figure 10 illustrates the results of VQGAN-LC with LDM using 1024 ($32 \times 32$) tokens on ImageNet. Figure 11 shows the unconditional generation results at a resolution of $256 \times 256$ for VQGAN-LC with LDM on the FFHQ dataset, utilizing 256 ($16 \times 16$) tokens. The introduction of a large-scale codebook facilitates the generation of images with diverse poses, intricate textures, and complex backgrounds.

## D  Limitations and Broad Impacts

We present a new image quantization technique called VQGAN-LC, which expands the codebook size to 100,000 with a utilization rate of 99%. This approach has the potential to enhance any downstream applications that involve image quantization models. However, VQGAN-LC is trained on the ImageNet and FFHQ datasets, which restricts downstream applications, like image generation, to producing images from the limited categories found in these datasets. While training VQGAN-LC on larger datasets like LAION-5B may improve its utility in downstream applications, it would also be more costly and computationally demanding. Furthermore, please refrain from using this model to generate malicious or inappropriate content. It is intended solely for positive and constructive purposes in research and creativity.

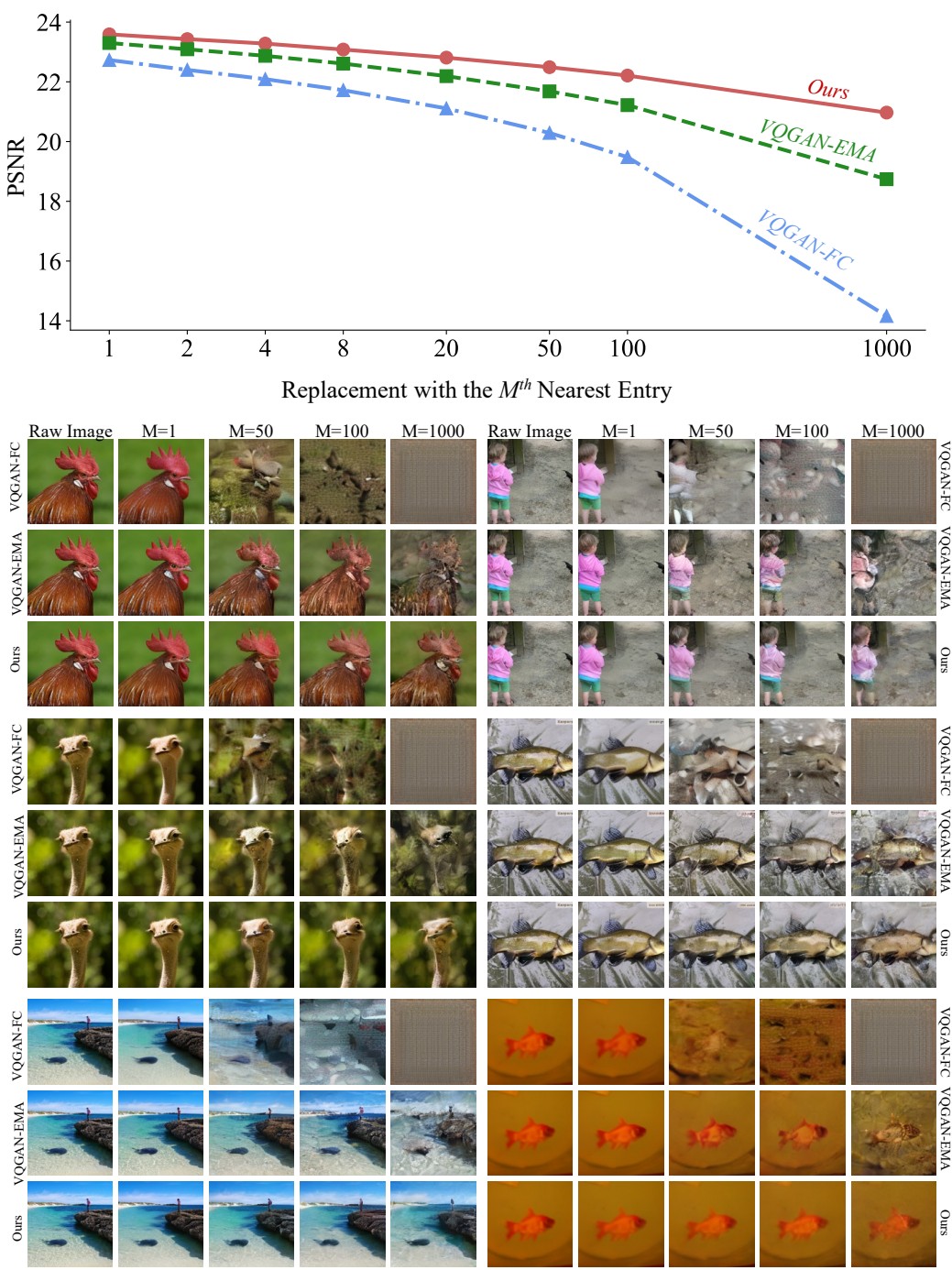

Figure 5: For a given image, we employ an image quantization model (VQGAN-FC, VQGAN-EMA, or our VQGAN-LC) to transform it into a token map. Each token in this map is then substituted with the $M^{th}$ nearest entry from the codebook. This altered token map is subsequently fed into the decoder for reconstruction. (Top) PSNR for each configuration. (Bottom) Reconstruction visualizations for the three models.

Australian terrier (193)      Rapeseed (984)      Agaric (992)

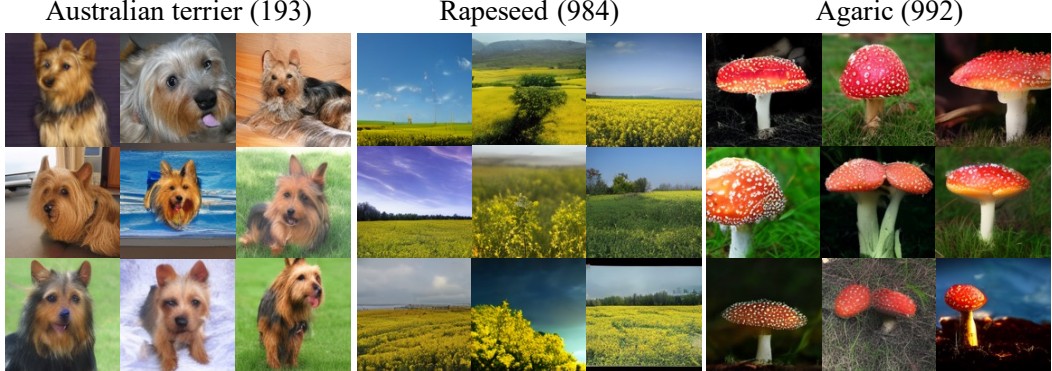

Figure 6: Qualitative results of class-conditional generation using our VQGAN-LC with GPT [2] on ImageNet, utilizing 256 ($16 \times 16$) tokens. We display the category name and corresponding category ID for each group.

Volcano (980)      Arctic Fox (279)      Pug (254)

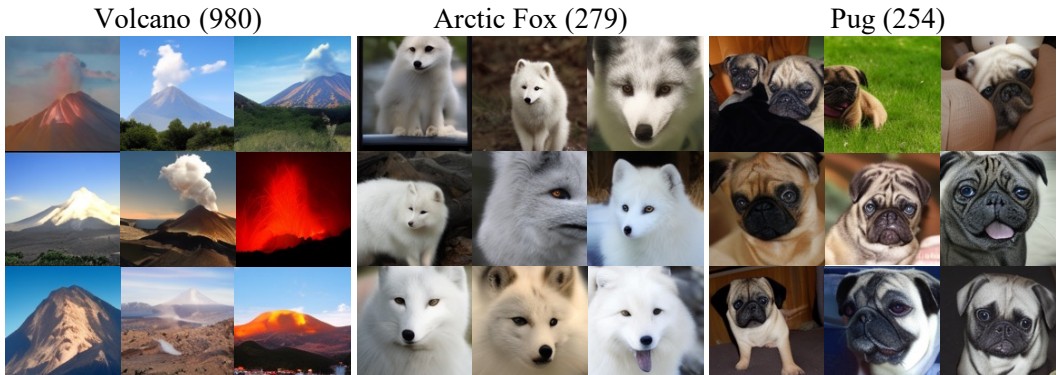

Figure 7: Qualitative results of class-conditional generation using our VQGAN-LC with LDM [3] on ImageNet, utilizing 256 ($16 \times 16$) tokens and a classifier-free guidance scale of 1.4. We display the category name and corresponding category ID for each group.

Lake Shore (975)      Geyser (974)      Space Shuttle (812)

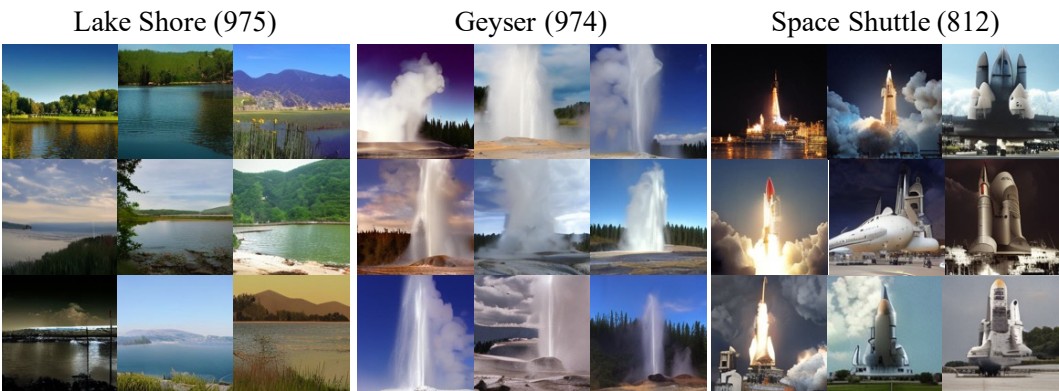

Figure 8: Qualitative results of class-conditional generation using our VQGAN-LC with DiT [4] on ImageNet, utilizing 256 ($16 \times 16$) tokens and a classifier-free guidance scale of 8.0. We display the category name and corresponding category ID for each group.

Cliff (972)          Balloon (417)          Guinea Pig (338)

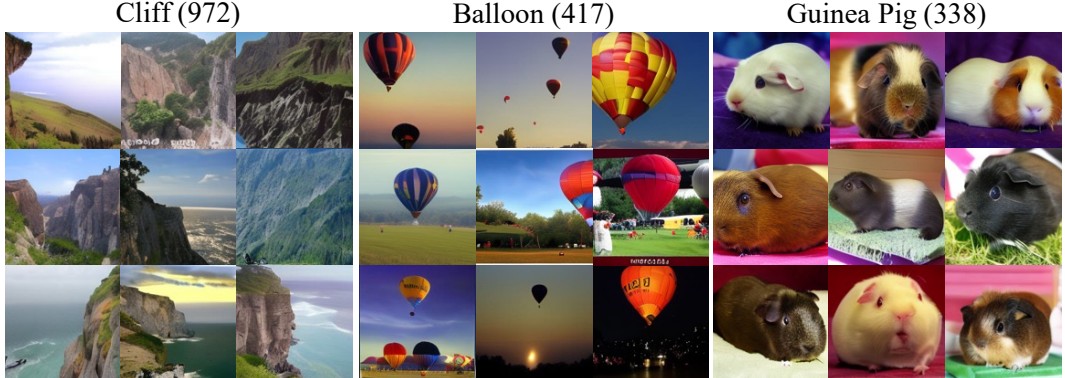

Figure 9: Qualitative results of class-conditional generation using our VQGAN-LC with SiT [5] on ImageNet, utilizing 256 (16 × 16) tokens and a classifier-free guidance scale of 8.0. We display the category name and corresponding category ID for each group.

Valley (979)          Red Panda (387)          Macaw (88)

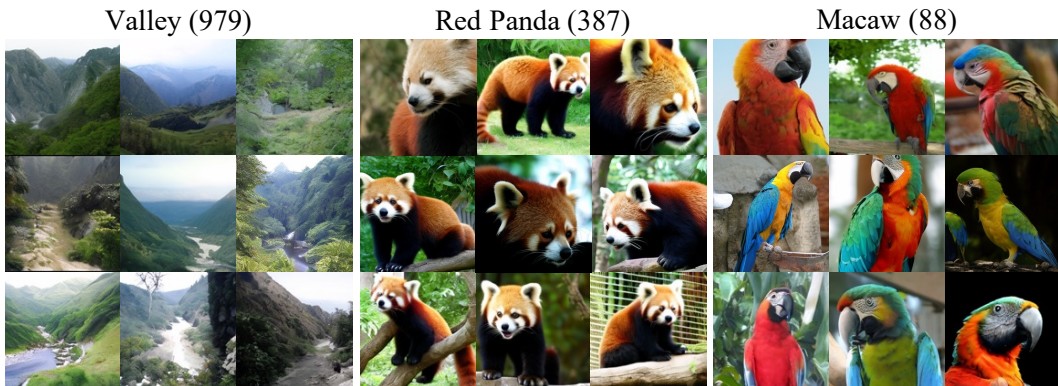

Figure 10: Qualitative results of class-conditional generation using our VQGAN-LC with LDM [3] on ImageNet, utilizing 1024 (32 × 32) tokens and a classifier-free guidance scale of 1.4. We display the category name and corresponding category ID for each group.

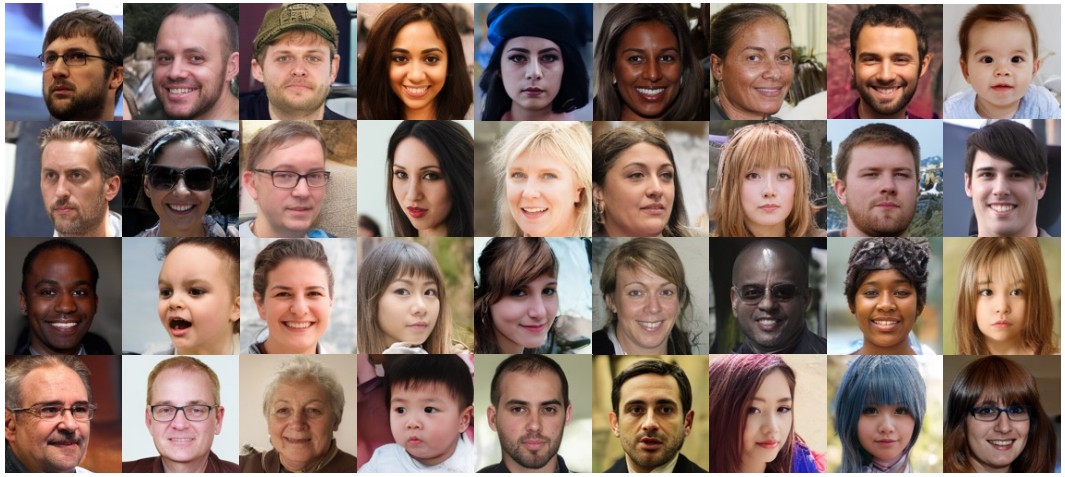

Figure 11: Qualitative results of unconditional generation using our VQGAN-LC with LDM [3] on FFHQ, utilizing 256 (16 × 16) tokens.

