# OpenReview forum: "Scaling the Codebook Size of VQ-GAN to 100,000 with a Utilization Rate of 99%"
_NeurIPS.cc/2024/Conference — NeurIPS 2024 poster_

### Official Review · Reviewer_igUX · 2024-07-11

**Soundness:** 2
**Presentation:** 3
**Contribution:** 2
**Rating:** 5
**Confidence:** 5

**Summary:**

This study introduces VQGAN-LC (Large Codebook), an innovative image quantization model that significantly extends the codebook size to 100,000, achieving a utilization rate of 99%. Unlike previous methods that optimize each codebook entry individually, VQGAN-LC initializes its codebook with 100,000 feature centers from a pre-trained vision encoder. These codebook vectors are then frozen for the remainder of the process. The optimization focuses on training a projector to align the frozen codebook with the feature distributions of the encoder in VQGAN-LC. This approach ensures that nearly all token embeddings remain active throughout the training phase.

**Strengths:**

(1)	Extensive number of experiments. Datasets and metrics are diverse.

(2)	Experiments are suitable for demonstrating the effectiveness of the proposed method.

(3)	Solid improvement in benchmarks.

**Weaknesses:**

(1)	The paper primarily focuses on scaling the codebook size and improving utilization rates in VQGAN models. While this is a noteworthy improvement, it can be seen as an incremental advance rather than a novel approach. The method relies heavily on existing architectures and concepts, particularly those established by prior works such as VQGAN, VQGAN-EMA, and VQGAN-FC.

(2)	The related work section fails to comprehensively cover the breadth of existing research focused on improving codebook usage and addressing the codebook collapse problem. Numerous significant contributions in this area are overlooked, which diminishes the depth and rigor of the literature review. This lack of thoroughness may lead to an incomplete understanding of the current state of the field and the novelty of the proposed approach. You may cite some recent papers for the codebook utilization problem, such as [1], [2], or other top papers that are found on arxiv by searching keywords: discrete VAEs codebook collapse.

(3)	There is no hypothesis to explain why the proposed method works. The paper does not address how the method increases the utilization rate and why this increase leads to improved results. This lack of theoretical foundation makes it difficult to understand the underlying mechanisms driving the observed performance gains.

(4)	The paper lacks experiments and explanations regarding the quality of the codebooks and how well they span the representational space. While the method claims high utility even on ResNet50, it is crucial to study the coverage and distribution of the codebook entries. Experiments should be conducted to demonstrate how the codebook spans the feature space and ensure that the high utilization rate translates to meaningful and diverse representations. Without such analysis, it is difficult to ascertain the true effectiveness and robustness of the proposed method.

(5)	In Preliminary D, a fixed step size (M=1, 50, 100, 1000) is used for the Mth nearest replacement. However, the ratio of the step size to the codebook size is not consistent. For example, if the codebook size is 16,000 in the baseline methods and a step size of M=1000 is used, it would be more meaningful to compare with a distance of 100,000/16 when the codebook size is 100,000. This is because, relatively, in VQGAN-FC, the replacement is made with a vector that is 1 step away in the codebook, whereas in the baseline methods, the replacement is made with a vector that is 6 steps farther away.

[1] Huh, M., Cheung, B., Agrawal, P., & Isola, P. (2023). Straightening Out the Straight-Through Estimator: Overcoming Optimization Challenges in Vector Quantized Networks. International Conference on Machine Learning.

[2] Takida, Y., Ikemiya, Y., Shibuya, T., Shimada, K., Choi, W., Lai, C.-H., … Mitsufuji, Y. (2024). HQ-VAE: Hierarchical Discrete Representation Learning with Variational Bayes. Transactions on Machine Learning Research

**Questions:**

(1)	When the codebook size increases significantly, I think that discretization might lose its meaning as the intuition behind discrete representation learning can be thought of as learning to represent the data space with a latent space as small as possible for a lower computational load. I’m curious about your justification for why we should increase the number of representations in the codebook instead of decreasing them, apart from better reconstruction performance.

(2)	One of my questions pertains to the span of the codebooks. Can you conduct a study or experiment to illustrate the coverage and distribution of the codebook entries? For instance, creating a t-SNE map by normalizing all codebooks and visualizing them in the same space could provide valuable insights into the codebook feature space learned by your model. This visualization would help us understand the diversity and quality of the representations within the codebook and how effectively it spans the feature space.

**Limitations:**

The authors address limitations and societal impact in Section D in Supplementary Material.

---

> ### Author Rebuttal · Authors · 2024-08-07
>
> Dear Reviewer igUX,
>
> Thanks for your valuable comments.
>
> **Q1: Dependence on Established Architectures**
>
> VQGAN and VQVAE are foundational works that introduced the encoder-quantizer-decoder framework for image quantization. Subsequent works, such as RQ-VAE, SQ-VAE, Reg-VQ, ViT-VQGAN, and the two references [1,2] you mentioned, adhere to this encoder-quantizer-decoder architecture without significant alterations. These works primarily focus on learning a discrete representation space that closely approximates the pixel space while minimizing information loss. To this end, they implement various improvements, such as: 1) increasing representation capability through multiple tokens for each image patch or replacing the traditional CNN backbone with a ViT backbone; 2) enhancing codebook utilization via stochastic quantization or reinitializing inactive codes during training.
>
> This paper builds upon the encoder-quantizer-decoder architecture established by VQGAN and VQVAE, similar to most previous works. However, existing research has not investigated the potential of learning an extremely large codebook while maintaining high codebook utilization. We undertake pioneering efforts to explore this possibility. Our experiments demonstrate that our VQGAN-LC can achieve this, establishing a novel approach for learning a robust discrete representation space that closely approximates the pixel space with minimal information loss.
>
> Please see **Q6** in the **global text response** for the significance of increasing the codebook size.
>
> **Q2: Discussion of Recent Papers**
>
> Thank you for highlighting these two excellent papers [1,2]. After a thorough review, we have identified several commonalities between the works we have discussed (CVQ-VAE and RegVQ) and the papers [1,2] you mentioned: 1) Both CVQ-VAE and [1] adopts the concept of online clustering to enhance codebook utilization; 2) Both RegVQ and [2] use stochastic quantization to prevent codebook collapse. We will incorporate a detailed discussion of the works [1-2] you mentioned in our revised version.
>
> **Q3: Explanation of the Motivations and Underlying Mechanisms**
>
> In L52-61, we examine why prior VQGAN variants struggle to achieve a high codebook utilization rate when the codebook size is significantly increased and the drawbacks of random codebook initialization. This setup leads to only a small portion of codes being optimized in each iteration. This observation inspired us to develop VQGAN-LC, which, instead of optimizing a small set of codes per iteration, initializes the codebook with a pre-trained visual encoder and optimizes the entire codebook distribution directly, as detailed in L62-72. In L146-152 and Figure 3, we analyze the codebook utilization rate throughout the training epochs for VQGAN-FC, VQGAN-EMA, and our VQGAN-LC, and the models' average utilization frequency across all epochs. Figure 4 presents t-SNE maps of the codebook used in our model and the baseline models. This analysis demonstrates that, during the training phase of image quantization models, prior works like VQGAN-FC and VQGAN-EMA have a significant number of codebook entries that do not receive any supervision signals, resulting in suboptimal representation capabilities.
>
> For further discussion, please see **Q6** in the **global text response**.
>
> **Q4: Codebook Coverage of Representational Space and T-SNE Visualization**
>
> Figure 4 in the main paper (Appendix B) showcases the active and inactive codes from the codebook for three models (VQGAN-FC, VQGAN-EMA, and our VQGAN-LC) at different codebook sizes (1,024, 16,384, 50,000, and 100,000), using t-SNE maps. Active codes are those that contribute to converting images into token maps, thereby defining the discrete representational space. Compared to the baseline models (VQGAN-FC and VQGAN-EMA), our method successfully expands the codebook size to 100,000 while maintaining a 99\% utilization rate, resulting in a significantly large number of active codes.
>
> Additionally, in **Figure 1 of the global rebuttal PDF file**, we provide t-SNE visualizations of the active codes from the codebook for the three models (VQGAN-FC, VQGAN-EMA, and our VQGAN-LC) as well as a combined t-SNE visualization for these models in the same space.
>
> **Q5: The M-th Nearest Replacement**
>
> We conjecture that you may refer to Figure 4 in the main paper (Appendix B) rather than "Preliminary D". To address your concerns, we performed the M-th nearest replacement on three models: two baselines, VQGAN-FC and VQGAN-EMA, and our model, VQGAN-LC. Each model uses a codebook of the same size, 100,000, to ensure a fair comparison. Please see **Figure 2 in the global rebuttal PDF file** for the results. This experiment illustrates that learning a large codebook with an extremely high code utilization rate enhances the representation capability and provides finer-grained representations within the image quantization model.
>
> **Q6: Why Increase the Number of Representations in the Codebook**
>
> The comprehensive answer to this question can be found in the **global text response**.

---

> > ### Comment · Reviewer_igUX · 2024-08-12
> > **Response to rebuttal**
> >
> > I have read the rebuttal. The authors’ responses are mostly satisfactory, particularly with addition of the new experiments. However, I still believe this work still lacks a theoretical foundation. It is a strong experimental paper.
> > I will increase my score accordingly.

---

### Official Review · Reviewer_c7tj · 2024-07-12

**Soundness:** 3
**Presentation:** 3
**Contribution:** 3
**Rating:** 7
**Confidence:** 3

**Summary:**

The paper introduces VQGAN-LC (Large Codebook), a novel image quantization model that significantly extends the codebook size and enhances codebook utilization. Traditional models like VQGAN-FC are limited in codebook size and utilization rates, with a maximum size of 16,384 and utilization rates typically below 12%. In contrast, VQGAN-LC expands the codebook size to 100,000 and achieves utilization rates exceeding 99%. Instead of optimizing each codebook entry individually, the model trains a projector that aligns the entire codebook with the encoder's feature distributions. This approach allows for a significant increase in codebook size and utilization without a corresponding increase in computational cost.

**Strengths:**

The most important contribution of this work is a trainable projector that maps the codebook entries to a latent space. The approach ensures nearly full codebook utilization throughout the training process, allowing for the codebook size to expand to over 100,000 entries while maintaining an impressive utilization rate of 99%.

The proposed method can be easily incorporated into existing VQGAN architecture.

The ability to scale the codebook size to up to 100,000 entries without incurring significant additional computational cost is critical.

**Weaknesses:**

In line 211 of page 6 ​​authors mention that the optimal codebook size of the proposed method is 100,000 which is intuitively correct. However, authors should also provide a comparison of smaller codebook sizes with other methods. In Table 3, Table 4, and Table 5  there is no comparison of VQGAN-LC with other methods utilizing a similar codebook size.

Quantization error constitutes a principal component of the loss function equation; however, in the two variants, Factorized Codes (FC) and Exponential Moving Average (EMA), distinct formulations of this loss are observed. The rationale provided may be comprehensible to those familiar with the VQGAN architecture; nonetheless, the explanations offered in the document lack sufficient clarity for those less acquainted with this framework.

Misc:
Mention the batch size for the experiments in the experimental setup section

**Questions:**

How does  Utilization Rate effect the performance? There seems to be a strong correlation for large codebook but for codebook sized 16,384 with “83% and 68%” Utilization achieve similar results compared to 99%Utilization.  In some cases lower Utilization has slightly better performance as well?

how does increasing codebook effect the inference time and memory footprint of the network. The paper claims in section 4.3 line#241 that large codebook incurs almost no additional computational cost. This is not obvious. How is computational cost independent of the codebook size?

Can you explain more about the perceptual loss used in eq.(1)

Does high utilization rate is any relationship with compressibility of the model? Intuitively lower utilization rates suggest that the model can be further compressed/ optimized? Will high utilization rates mean that it decreases the compressibility of the model?

---

> ### Author Rebuttal · Authors · 2024-08-07
>
> Dear Reviewer c7tj,
>
> Thanks for your valuable comments.
>
> **Q1: Comparison of Smaller Codebook Sizes with Other Methods**
>
> Table 1 in the main paper compares our VQGAN-LC with baseline models VQGAN-FC and VQGAN-EMA across various codebook sizes (**1,024**, **16,384**, **50K**, and **100K**). The evaluation encompasses both reconstruction and generation using the latent diffusion model (LDM) on the ImageNet dataset. The results show that VQGAN-FC and VQGAN-EMA perform optimally at a codebook size of 16,384. In contrast, our VQGAN-LC supports codebook sizes up to 100,000 while maintaining an impressive utilization rate of 99\%.
>
> Based on your suggestions, we conducted additional experiments with our VQGAN-LC using a smaller codebook size of 16,384. All other configurations remained the same as in the main paper (e.g., representing an image with 256 tokens and an input image resolution of $256 \times 256$). We compared our model with baseline models and report: 1) reconstruction performance on ImageNet and FFHQ, using metrics such as rFID, LPIPS, PSNR, and SSIM (**the first table in global text response**); 2) image generation results on ImageNet, using various generation models including GPT, LDM, SiT, and DiT, with FID as the evaluation metric (**the second table in global text response**). Despite utilizing a smaller codebook of size 16,384, our VQGAN-LC surpasses the baseline models in both reconstruction and generation tasks. This experiment will be incorporated into our revised version.
>
> **The two tables can be found in the global text response.**
>
> **Q2: More Explanations for VQGAN-FC and VQGAN-EMA**
>
> Thank you for the suggestions. Due to the rebuttal's length constraints, we regret that we cannot provide further details at this time. However, we will include more comprehensive explanations of the VQGAN-FC and VQGAN-EMA formulations in our revised version.
>
> **Q3: Batch Sizes**
>
> The specifics of the batch sizes are provided in Appendix A of the main paper. Specifically, the batch size for training our VQGAN-LC is 256. For the training of GPT, LDM, DiT, and SiT, the batch sizes are 1024, 448, 256, and 256, respectively.
>
> **Q4: Impact of Utilization Rate on Performance**
>
> The performance of an image quantization model is determined by the number of active codes, which is the product of the utilization rate and the codebook size. Active codes are the ones that participate in converting images into token maps, thus defining the discrete representation space. For example, a model with a codebook size of 1024 and a utilization rate of 99\% has $1024 \times 0.99 = 1014$ active codes. In contrast, a model with a codebook size of 16,384 and a utilization rate of 83\% has $16,384 \times 0.83 = 13,599$ active codes. Although the first model has a higher utilization rate, its performance may be inferior to the latter model due to the smaller number of active codes. The superior performance of our VQGAN-LC model can be attributed to its ability to expand the codebook size up to 100,000 while maintaining a high utilization rate of 99\%. This high efficiency ensures that almost no codes are wasted, leading to improved training efficiency.
>
> **Q5: Inference and Memory Costs**
>
> All VQGAN models, including our VQGAN-LC and its variants, follow an encoder-quantizer-decoder architecture. In this work, we utilize the same encoder and decoder across all models, including ours. Let $F \in \mathcal{R}^{16 \times 16 \times C}$ represent the feature generated by the encoder, where $C$ is the feature dimension. Let $B \in \mathcal{R}^{N \times C}$ represent the codebook with size $N$. The quantizer performs a matrix multiplication between $F$ and $B$ to transform $F$ into a token map, where each element corresponds to an entry in the codebook. The primary inference cost is attributed to the encoder and decoder. Increasing the codebook size $N$ incurs minimal additional cost since the matrix multiplication between $F$ and $B$ is negligible compared to the encoder and decoder processing time. Below, we present the multiply-accumulates (MACs) and model sizes for our VQGAN-LC models with codebook sizes of 16,384 and 100,000, respectively, when inferring an image of size $256 \times 256$.
>
> | Codebook Size | MACs   | Model Size |
> |---------------|--------|------------|
> | 16,384        | 195.08G| 71.71M     |
> | 100,000       | 195.70G| 71.72M     |
>
> **Q6: Perceptual Loss**
>
> Perceptual loss is widely used in image generation. Unlike the traditional reconstruction loss, which calculates the mean squared error (MSE) directly between the raw and reconstructed images in the pixel space, perceptual loss computes the MSE between the feature representations of the raw and reconstructed images. These feature representations are extracted using a pre-trained VGG network, and the loss is applied in the feature space rather than the pixel space. We will provide further details about the perceptual loss in our revision.
>
> **Q7: Relationship between Utilization Rate and Compressibility**
>
> In image quantization models designed for generative purposes, the primary aim is to convert images into discrete tokens with minimal information loss, enabling the generation of diverse and realistic images. As discussed in **Q5**, the VQGAN family primarily focuses on the number of active codes in the codebook. Typically, an image is represented by a fixed set of tokens across all VQGAN models, including ours, resulting in a constant level of compressibility. However, the number of active codes determines the span of the discrete space. A higher number of active codes indicates a more powerful representation capability and finer-grained representations within the image quantization model. This, in turn, can lead to the generation of more diverse and realistic images for downstream generative models such as GPT, LDM, SiT, and DiT. We will incorporate these discussions into our revision.

---

> > ### Comment · Reviewer_c7tj · 2024-08-12
> >
> > I have read the rebuttal and I am satisfied with the Authors response.
> >
> > However I have one additional question: in the main rebuttal authors have mentioned
> >
> > "The primary objective of GPT-style image generation using image quantization models is to produce high-quality, diverse, and realistic images, rather than minimizing the size of the codebook used to represent images."
> >
> > Can you please provide any references for this statement

---

> ### Author Response · Authors · 2024-08-12
> **Response to Reviewer c7tj**
>
> Dear Reviewer c7tj,
>
> Thank you for your question. We appreciate the opportunity to clarify this point.
>
> In our rebuttal, we highlighted that the main goal in GPT-style image generation using image quantization models is to produce high-quality, diverse, and realistic images. Achieving this often involves expanding the codebook (increasing its size or employing better optimization strategies), using more tokens to represent an image (e.g., from 256 to 1024 tokens), or adopting a more advanced backbone (e.g., replacing CNN with ViT).
>
> Several statements from vector quantization models [1][2][3] support our argument. VQGAN [1], VQ-VAE-2 [2], and Reg-VQ [3] all follow a two-step process for image generation: first, an image quantization model is trained to convert an image into a sequence of tokens, and then a GPT is trained to model these discrete tokens. These statements include:
>
> - In VQGAN [1], the authors state, "using **transformers** to represent images as a distribution over latent image constituents requires us to push the limits of compression and **learn a rich codebook**."
>
> - In VQ-VAE-2 [2], the authors state, "we **scale and enhance the autoregressive priors** used in VQ-VAE to generate synthetic samples of **much higher coherence and fidelity** than possible before."
>
> - In Reg-VQ [3], the authors state, "we can observe that **the performance of regularized quantization (Reg-VQ) improves clearly** with **the increasing of codebook size**."
>
>
> A consistent theme across these works is **their emphasis on enhancing the quality of the generated images**. They adopt different strategies, such as expanding the codebook size or increasing the number of tokens used for image representation, to achieve this objective. It is important to highlight, as discussed in response to "Q5: Inference and Memory Costs", that expanding the codebook size results in minimal additional costs, while increasing the number of tokens used to represent an image substantially raises the GPT training and inference expenses.
>
>
> Thank you once again. We look forward to continuing our discussions with you.
>
>
> [1] Taming Transformers for High-Resolution Image Synthesis, CVPR 2021.
>
> [2] Generating Diverse High-Fidelity Images with VQ-VAE-2, NeurIPS 2019.
>
> [3] Regularized Vector Quantization for Tokenized Image Synthesis, CVPR 2023.

---

> > ### Comment · Reviewer_c7tj · 2024-08-13
> >
> > Thank you for the detailed response.
> >
> > I am satisfied with the authors response and I have increased my score accordingly.

---

### Official Review · Reviewer_Kj4D · 2024-07-13

**Soundness:** 3
**Presentation:** 3
**Contribution:** 3
**Rating:** 6
**Confidence:** 4

**Summary:**

The VQGAN-LC (Large Codebook) model tackles the challenges of expanding codebook size and utilization in image quantization. Unlike its predecessors, which struggled with limited codebook sizes and low utilization rates, VQGAN-LC increases the codebook size to 100,000 and achieves over 99% utilization. This model starts with features extracted by a pre-trained vision encoder and optimizes a projector to align these features with the encoder's distributions. VQGAN-LC demonstrates superior performance in tasks such as image reconstruction, classification, and generative image models compared to earlier methods like VQGAN-FC and VQGAN-EMA.

**Strengths:**

1. This paper aims to address a very important topic, which may be generalized to other modalities, such as speech and video.
2. The structure and presentation of this paper is straightforward and easy to understand.
3. The experiments are comprehensive and persuasive, which shows obvious improvement over other SOTA methods.

**Weaknesses:**

1. My main concern is doubt about the scalability of this method. It appears that this paper does not provide enough evidence to show that it can still be useful for open-domain image reconstruction and generation under billion-level image datasets, since all experiments are carried out on small-scale datasets and low-resolution images (256x256).

**Questions:**

1. What is the definition of active or inactive for the utilization of the codebook? How to mesure it?
2. Can the tokenizer be integrated into large language models (GPT-like) to train a multimodal large language model?

**Limitations:**

The main limitation is also mentioned in the weakness.

---

> ### Author Rebuttal · Authors · 2024-08-07
>
> Dear Reviewer Kj4D,
>
> Thanks for your valuable comments.
>
> **Q1: Reconstruction and Generation on Billion-Level Datasets**
>
> - Firstly, we want to highlight that we adhere to the methodologies established in previous works such as VQGAN [1] and RQTransformer [2], conducting our experiments on the commonly used benchmarks ImageNet-1K and FFHQ for fair comparison. Training a standard VQGAN and our VQGAN-LC on ImageNet-1K, which consists of 1.28M images, requires 66 and 72 hours, respectively, using 32 V100 GPUs. Training a VQGAN or its variants on a significantly larger dataset like LAION-400M, which is nearly 300 times the size of ImageNet, could take up to 19,800 hours (825 days) with the same resources. This extensive training duration is prohibitively expensive, and we hope the reviewer understands the heavy training cost involved. Most previous studies validate their methods on ImageNet-1K. Only a few works can afford training on exceptionally large datasets. For example, Muse [3] trained a VQGAN-based generation model on LAION-400M, which took over a week using a 512-core TPU-v4.
>
> - However, to verify the scalability of our method within the limited rebuttal period, we train our VQGAN-LC on a combination of ImageNet-1K, which includes 1.28M images, and a subset (termed LAION-1M) containing an equivalent number of images sampled from LAION-400M. We report reconstruction performance using the rFID metric in the following table. Our approach demonstrates the benefits of data scalability: training on larger datasets significantly reduces the rFID scores across validation sets of various benchmarks. We will further verify large-scale training in our revision.
>
> | Dataset                   | Codebook Size | Utilization (\%) | ImageNet (val) | LAION (val) | FFHQ (val) |
> |------|------|--------|---------|--------|--------|
> | ImageNet        | 100,000      | 99.4 | 2.62          | 5.73    | 7.29        |
> | ImageNet + LAION-1M | 100,000  | **99.6** |**2.36**      | **3.87**     | **6.86**      |
>
> **Q2: Experiments on High-Resolution Images**
>
> Below, we present the reconstruction performance on ImageNet, which contains images with a resolution of $512 \times 512$ pixels. Without altering the network structure, each image is quantized into a $32 \times 32$ token map (i.e. 1024 tokens). It is worth noting that the optimal codebook size for the baseline models, VQGAN-FC and VQGAN-EMA, remains 16,384, whereas for our VQGAN-LC, it is 100,000. Our model consistently surpasses all baselines in high-resolution settings, as evidenced by various metrics such as rFID, LPIPS, PSNR, and SSIM. We will include this experiment in our revised version.
>
> | Method       | # Tokens | Codebook Size | Utilization (\%) | rFID | LPIPS | PSNR | SSIM |
> |---------|-----|---------|--------|----|-------|------|------|
> | VQGAN-FC     | 1024     | 16,384        | 11.1             | 2.15 | 0.13  | 25.8 | 72.8 |
> | VQGAN-EMA    | 1024     | 16,384        | 85.3             | 1.76 | 0.12  | 26.6 | 74.4 |
> | VQGAN-LC (Ours) | 1024 | 100,000       | **99.7**         | **1.51** | **0.11** | **27.1** | **77.4** |
>
> **Q3: Defining and Measuring Active and Inactive Tokens**
>
> VQGAN, its variants, and our VQGAN-LC use a codebook for image quantization. These models convert each image into a token map, with each token corresponding to a codebook entry. After training, we quantize all images from the training set into token maps. Codebook entries that are never used, often due to suboptimal training strategies, are designated as inactive tokens (codes). In contrast, codebook entries used at least once to represent an image in the training set are classified as active tokens (codes).
>
> The codebook utilization rate is calculated as the ratio of active entries (tokens/codes) to the total size of the codebook. Figure 3 illustrates the codebook utilization rate across training epochs and the average utilization frequency. Figure 4 visualizes active and inactive tokens using t-SNE.
>
> **Q4: Integrating this Work into GPT-like LLMs for Training Multimodal LLMs**
>
> This is an excellent suggestion! GPT-like autoregressive models can predict subsequent tokens using a causal Transformer decoder. By integrating an image quantization model, such as our VQGAN-LC, we can enable large multimodal models to generate both language and image tokens. These generated image tokens are then processed by the image quantization model's decoder to produce realistic images.
>
> On the one hand, some existing works have already explored this direction by utilizing an image quantization model to develop multimodal LLMs. For instance, VideoPoet [4] employs MAGVIT-v2 [5] as the quantization model to generate content in both language and visual modalities. However, training VideoPoet requires extensive resources, and due to limited rebuttal time and computational resources, we were unable to conduct such experiments.
>
> On the other hand, our work, like prior research on image quantization, aims to develop an improved quantization model. Image understanding and generation are central to multimodal LLMs. We compare our VQGAN-LC with several models in both understanding and generation tasks. In Figure 1.b, we demonstrate the image classification for the understanding task. In Tables 4 and 5, we showcase the image generation capability by applying our VQGAN-LC to various generation models, including autoregressive image generation (GPT), and diffusion- and flow-based generative models (LDM, DiT, and SiT).
>
> We will include these discussions in our revision.
>
> **Reference**
>
> [1] Taming transformers for high-resolution image synthesis, CVPR 2021.
>
> [2] Autoregressive image generation using residual quantization, CVPR 2022.
>
> [3] Muse: Text-to-image generation via masked generative transformers, ICML 2023.
>
> [4] VideoPoet: A Large Language Model for Zero-Shot Video Generation, ICML 2024.
>
> [5] Language Model Beats Diffusion-Tokenizer is Key to Visual Generation, ICLR 2024.

---

### Author Rebuttal · Authors · 2024-08-07

We thank all reviewers for their constructive comments.

In this global response, we provide the performance tables for **Reviewer c7tj**, and address some comments from **Reviewer igUX**.

**Reviewer-c7tj-Q1: Comparison of Smaller Codebook Sizes with Other Methods**

Reconstruction performance on FFHQ and ImageNet:

| Method             | # Tokens | Codebook Size | Utilization (\%) | rFID | LPIPS | PSNR | SSIM |
|-|-|-|-|-|-|-|-|
| **FFHQ**   |          |               |                  |      |       |      |      |
| VQGAN              | 256      | 16,384        | 2.3              | 5.25 | 0.12  | 24.4 | 63.3 |
| VQGAN-FC           | 256      | 16,384        | 10.9             | 4.86 | 0.11  | 24.8 | 64.6 |
| VQGAN-EMA          | 256      | 16,384        | 68.2             | 4.79 | 0.10  | 25.4 | 66.1 |
| VQGAN-LC (Ours)    | 256      | 16,384        | 99.9         | 4.22 | 0.09  | 25.7 | 68.0 |
| VQGAN-LC (Ours)    | 256      | 100,000       | 99.5             | **3.81** | **0.08** | **26.1** | **69.4** |
| **ImageNet** |        |               |                  |      |       |      |      |
| VQGAN              | 256      | 16,384        | 3.4              | 5.96 | 0.17  | 23.3 | 52.4 |
| VQGAN-FC           | 256      | 16,384        | 11.2             | 4.29 | 0.17  | 22.8 | 54.5 |
| VQGAN-EMA          | 256      | 16,384        | 83.2             | 3.41 | 0.14  | 23.5 | 56.6 |
| VQGAN-LC (Ours)    | 256      | 16,384        | 99.9         | 3.01 | 0.13  | 23.2 | 56.4 |
| VQGAN-LC (Ours)    | 256      | 100,000       | 99.9         | **2.62** | **0.12** | **23.8** | **58.9** |

Generation results on ImageNet:

| Method             | # Tokens | Codebook Size  | GPT | SiT | DiT |  LDM |
|-|-|-|-|-|-|-|
| VQGAN-FC           | 256      | 16,384         | 17.3   | 10.3   | 13.7   | 9.78   |
| VQGAN-EMA          | 256      | 16,384         | 16.3   | 9.31   | 13.4   | 9.13   |
| VQGAN-LC (Ours)    | 256      | 16,384         | 16.1   | 9.06   | 11.2  | 8.84   |
| VQGAN-LC (Ours)    | 256      | 100,000        | **15.4**   | **8.40**   | **10.8**   | **8.36**   |

**Reviewer-igUX-Q6: Why Increase the Number of Representations in the Codebook**

In light of GPT's notable success in auto-regressive language generation, image quantization models, such as VQGAN, have been developed to facilitate GPT-style image generation. Given the inherently continuous nature of image signals, directly applying GPT to image generation presents substantial challenges. To address this, image quantization techniques have been introduced, effectively transforming images into token maps. This conversion enables the training of GPT models to generate images by predicting flattened token maps in an auto-regressive manner. This approach not only leverages the strengths of GPT in handling sequential data but also facilitates the efficient and coherent synthesis of high-quality images, bridging the gap between language and visual data processing.

The primary objective of GPT-style image generation using image quantization models is to produce high-quality, diverse, and realistic images, rather than minimizing the size of the codebook used to represent images. In our experiments, we discovered that training our VQGAN-LC with a codebook size of 100,000 on ImageNet incurs only a 1\% additional cost compared to training a model with a codebook size of 16,384. The extra inference cost for converting an image into a token map is negligible, adding just less than 1\% to the total cost. Training a GPT model (480M) using a codebook size of 100,000 incurs only 1.3\% cost than using one with a codebook size of 16,384. Below, we present the multiply-accumulates (MACs) and model sizes for our VQGAN-LC models with codebook sizes of 16,384 and 100,000, respectively, when inferring an image of size $256 \times 256$.

| Codebook Size | MACs   | Model Size |
|-|-|-|
| 16,384        | 195.08G| 71.71M     |
| 100,000       | 195.70G| 71.72M     |

Previous baseline models, such as VQGAN-FC and VQGAN-EMA, have shown that increasing the codebook size significantly improves performance. For example, **the original VQGAN demonstrated that expanding the codebook size from 1,024 to 16,384 reduced the reconstruction FID from 7.94 to 4.98.** However, these VQGAN variants have not explored the utilization of a codebook larger than 16,384. As illustrated in Figure 1 of the main paper, our re-implementation of two VQGAN baselines indicates that these models do not benefit from a codebook larger than 16,384 using their codebook optimization strategies.

In this work, we successfully expand the codebook size to 100,000 while maintaining a codebook utilization rate of over 99\%. In Figure 1 and Tables 2-5, our VQGAN-LC and all baseline models utilize the same encoder-decoder network and generation models (such as GPT, LDM, SiT, and DiT), differing only in codebook size and optimization strategy. The optimal codebook size for VQGAN-FC and VQGAN-EMA is 16,384, while for our VQGAN-LC, it is 100,000. Our model consistently outperforms the baseline models in image classification, image reconstruction, and image generation using various generative models, highlighting the potential of a larger, well-optimized codebook. This is because a larger codebook allows for a more detailed representation of the input data, capturing subtle variations and intricate details that smaller codebooks may miss.

The improvement in GPT generation performance with an increased codebook size is not unique to image generation. Studies on LLMs suggest that employing a tokenizer with an expanded vocabulary significantly enhances model efficacy. For example, the technical report (https://ai.meta.com/blog/meta-llama-3/) for LLAMA 3 shows, "LLAMA 3 uses a tokenizer with a vocabulary of 128K tokens that encodes language much more efficiently, which leads to substantially improved model performance."

We will include these discussions in our revision.

---

### Decision · Program_Chairs · 2024-09-25

**Decision:**

Accept (poster)

**Comment:**

All reviewers suggested acceptance, altough without much enthusiasm. The paper's contribution is more of an engineering effort, but the practical results are good.